# An Inexact Augmented Lagrangian Framework for Nonconvex Optimization with Nonlinear Constraints

**Mehmet Fatih Sahin**
mehmet.sahin@epfl.ch

**Armin Eftekhari**
armin.eftekhari@epfl.ch

**Ahmet Alacaoglu**
ahmet.alacaoglu@epfl.ch

**Fabian Latorre**
fabian.latorre@epfl.ch

**Volkan Cevher**
volkan.cevher@epfl.ch

LIONS, Ecole Polytechnique Fédérale de Lausanne, Switzerland

## Abstract

We propose a practical inexact augmented Lagrangian method (iALM) for nonconvex problems with nonlinear constraints. We characterize the total computational complexity of our method subject to a verifiable geometric condition, which is closely related to the Polyak-Lojasiewicz and Mangasarian-Fromowitz conditions.

In particular, when a first-order solver is used for the inner iterates, we prove that iALM finds a first-order stationary point with $\tilde{\mathcal{O}}(1/\epsilon^3)$ calls to the first-order oracle. If, in addition, the problem is smooth and a second-order solver is used for the inner iterates, iALM finds a second-order stationary point with $\tilde{\mathcal{O}}(1/\epsilon^5)$ calls to the second-order oracle. These complexity results match the known theoretical results in the literature.

We also provide strong numerical evidence on large-scale machine learning problems, including the Burer-Monteiro factorization of semidefinite programs, and a novel nonconvex relaxation of the standard basis pursuit template. For these examples, we also show how to verify our geometric condition.

## 1 Introduction

We study the nonconvex optimization problem

$$\min_{x \in \mathbb{R}^d} f(x) + g(x) \quad \text{s.t.} \quad A(x) = 0, \tag{1}$$

where $f : \mathbb{R}^d \to \mathbb{R}$ is a continuously-differentiable nonconvex function and $A : \mathbb{R}^d \to \mathbb{R}^m$ is a nonlinear operator. We assume that $g : \mathbb{R}^d \to \mathbb{R} \cup \{\infty\}$ is a proximal-friendly convex function [47].

A host of problems in computer science [33, 37, 70], machine learning [40, 59], and signal processing [57, 58] naturally fall under the template (1), including max-cut, clustering, generalized eigenvalue decomposition, as well as the quadratic assignment problem (QAP) [70].

To solve (1), we propose an intuitive and easy-to-implement augmented Lagrangian algorithm, and provide its total iteration complexity under an interpretable geometric condition. Before we elaborate on the results, let us first motivate (1) with an application to semidefinite programming (SDP):

**Vignette: Burer-Monteiro splitting.** A powerful convex relaxation for max-cut, clustering, and many others is provided by the trace-constrained SDP

$$\min_{X \in \mathbb{S}^{d \times d}} \langle C, X \rangle \quad \text{s.t.} \quad B(X) = b, \ \text{tr}(X) \le \alpha, \ X \succeq 0, \tag{2}$$

where $C \in \mathbb{R}^{d \times d}$, $X$ is a positive semidefinite $d \times d$ matrix, and $B : \mathbb{S}^{d \times d} \to \mathbb{R}^m$ is a linear operator. If the unique-games conjecture is true, the SDP (2) obtains the best possible approximation for the underlying discrete problem [53].

Since $d$ is often large, many first- and second-order methods for solving such SDP's are immediately ruled out, not only due to their high computational complexity, but also due to their storage requirements, which are $\mathcal{O}(d^2)$.

A contemporary challenge in optimization is therefore to solve SDPs using little space and in a scalable fashion. The recent homotopy conditional gradient method, which is based on linear minimization oracles (LMOs), can solve (2) in a small space via sketching [69]. However, such LMO-based methods are extremely slow in obtaining accurate solutions.

A different approach for solving (2), dating back to [14, 15], is the so-called Burer-Monteiro (BM) factorization $X = UU^\top$, where $U \in \mathbb{R}^{d \times r}$ and $r$ is selected according to the guidelines in [49, 1], which is tight [63]. The BM factorization leads to the following nonconvex problem in the template (1):

$$\min_{U \in \mathbb{R}^{d \times r}} \langle C, UU^\top \rangle \quad \text{s.t.} \quad B(UU^\top) = b, \ \|U\|_F^2 \leq \alpha, \tag{3}$$

The BM factorization does not introduce any extraneous local minima [15]. Moreover, [13] establishes the connection between the local minimizers of the factorized problem (3) and the global minimizers for (2). To solve (3), the inexact Augmented Lagrangian method (iALM) is widely used [14, 15, 35], due to its cheap per iteration cost and its empirical success.

Every (outer) iteration of iALM calls a solver to solve an intermediate augmented Lagrangian subproblem to near stationarity. The choices include first-order methods, such as the proximal gradient descent [47], or second-order methods, such as the trust region method and BFGS [44].[1]

Unlike its convex counterpart [41, 36, 65], the convergence rate and the complexity of iALM for (3) are not well-understood, see Section 5 for a review of the related literature. Indeed, addressing this important theoretical gap is one of the contributions of our work. In addition,

▷ We derive the convergence rate of iALM to first-order optimality for solving (1) or second-order optimality for solving (1) with $g = 0$, and find the total iteration complexity of iALM using different solvers for the augmented Lagrangian subproblems. Our complexity bounds match the best theoretical results in optimization, see Section 5.

▷ Our iALM framework is future-proof in the sense that different subsolvers can be substituted.

▷ We propose a geometric condition that simplifies the algorithmic analysis for iALM, and clarify its connection to well-known Polyak-Lojasiewicz [32] and Mangasarian-Fromovitz [3] conditions. We also verify this condition for key problems in Appendices D and E.

## 2 Preliminaries

**Notation.** We use the notation $\langle \cdot, \cdot \rangle$ and $\| \cdot \|$ for the standard inner product and the norm on $\mathbb{R}^d$. For matrices, $\| \cdot \|$ and $\| \cdot \|_F$ denote the spectral and the Frobenius norms, respectively. For the convex function $g : \mathbb{R}^d \to \mathbb{R}$, the subdifferential set at $x \in \mathbb{R}^d$ is denoted by $\partial g(x)$ and we will occasionally use the notation $\partial g(x)/\beta = \{z/\beta : z \in \partial g(x)\}$. When presenting iteration complexity results, we often use $\widetilde{O}(\cdot)$ which suppresses the logarithmic dependencies.

We denote $\delta_{\mathcal{X}} : \mathbb{R}^d \to \mathbb{R}$ as the indicator function of a set $\mathcal{X} \subset \mathbb{R}^d$. The distance function from a point $x$ to $\mathcal{X}$ is denoted by $\text{dist}(x, \mathcal{X}) = \min_{z \in \mathcal{X}} \|x - z\|$. For integers $k_0 \leq k_1$, we use the notation $[k_0 : k_1] = \{k_0, \ldots, k_1\}$. For an operator $A : \mathbb{R}^d \to \mathbb{R}^m$ with components $\{A_i\}_{i=1}^m$, $DA(x) \in \mathbb{R}^{m \times d}$ denotes the Jacobian of $A$, where the $i$th row of $DA(x)$ is the vector $\nabla A_i(x) \in \mathbb{R}^d$.

**Smoothness.** We assume smooth $f : \mathbb{R}^d \to \mathbb{R}$ and $A : \mathbb{R}^d \to \mathbb{R}^m$; i.e., there exist $\lambda_f, \lambda_A \geq 0$ s.t.

$$\|\nabla f(x) - \nabla f(x')\| \leq \lambda_f \|x - x'\|, \quad \|DA(x) - DA(x')\| \leq \lambda_A \|x - x'\|, \quad \forall x, x' \in \mathbb{R}^d. \tag{4}$$

**Augmented Lagrangian method (ALM).** ALM is a classical algorithm, which first appeared in [29, 51] and extensively studied afterwards in [3, 8]. For solving (1), ALM suggests solving the

problem

$$\min_x \max_y \ \mathcal{L}_\beta(x, y) + g(x), \tag{5}$$

where, for penalty weight $\beta > 0$, $\mathcal{L}_\beta$ is the corresponding augmented Lagrangian, defined as

$$\mathcal{L}_\beta(x, y) := f(x) + \langle A(x), y \rangle + \frac{\beta}{2} \|A(x)\|^2. \tag{6}$$

The minimax formulation in (5) naturally suggests the following algorithm for solving (1):

$$x_{k+1} \in \underset{x}{\operatorname{argmin}} \ \mathcal{L}_\beta(x, y_k) + g(x), \tag{7}$$

$$y_{k+1} = y_k + \sigma_k A(x_{k+1}),$$

where the dual step sizes are denoted as $\{\sigma_k\}_k$. However, computing $x_{k+1}$ above requires solving the nonconvex problem (7) to optimality, which is typically intractable. Instead, it is often easier to find an approximate first- or second-order stationary point of (7).

Hence, we argue that by gradually improving the stationarity precision and increasing the penalty weight $\beta$ above, we can reach a stationary point of the main problem in (5), as detailed in Section 3.

**Optimality conditions.** First-order necessary optimality conditions for (1) are well-studied. Indeed, $x \in \mathbb{R}^d$ is a first-order stationary point of (1) if there exists $y \in \mathbb{R}^m$ such that

$$-\nabla_x \mathcal{L}_\beta(x, y) \in \partial g(x), \qquad A(x) = 0, \tag{8}$$

which is in turn the necessary optimality condition for (5). Inspired by this, we say that $x$ is an $(\epsilon_f, \beta)$ first-order stationary point of (5) if there exists a $y \in \mathbb{R}^m$ such that

$$\operatorname{dist}(-\nabla_x \mathcal{L}_\beta(x, y), \partial g(x)) \leq \epsilon_f, \qquad \|A(x)\| \leq \epsilon_f, \tag{9}$$

for $\epsilon_f \geq 0$. In light of (9), a metric for evaluating the stationarity of a pair $(x, y) \in \mathbb{R}^d \times \mathbb{R}^m$ is

$$\operatorname{dist}(-\nabla_x \mathcal{L}_\beta(x, y), \partial g(x)) + \|A(x)\|, \tag{10}$$

which we use as the first-order stopping criterion. As an example, for a convex set $\mathcal{X} \subset \mathbb{R}^d$, suppose that $g = \delta_{\mathcal{X}}$ is the indicator function on $\mathcal{X}$. Let also $T_{\mathcal{X}}(x) \subseteq \mathbb{R}^d$ denote the tangent cone to $\mathcal{X}$ at $x$, and with $P_{T_{\mathcal{X}}(x)} : \mathbb{R}^d \to \mathbb{R}^d$ we denote the orthogonal projection onto this tangent cone. Then, for $u \in \mathbb{R}^d$, it is not difficult to verify that

$$\operatorname{dist}(u, \partial g(x)) = \|P_{T_{\mathcal{X}}(x)}(u)\|. \tag{11}$$

When $g = 0$, a first-order stationary point $x \in \mathbb{R}^d$ of (1) is also second-order stationary if

$$\lambda_{\min}(\nabla_{xx} \mathcal{L}_\beta(x, y)) \geq 0, \tag{12}$$

where $\nabla_{xx} \mathcal{L}_\beta$ is the Hessian of $\mathcal{L}_\beta$ with respect to $x$, and $\lambda_{\min}(\cdot)$ returns the smallest eigenvalue of its argument. Analogously, $x$ is an $(\epsilon_f, \epsilon_s, \beta)$ second-order stationary point if, in addition to (9), it holds that

$$\lambda_{\min}(\nabla_{xx} \mathcal{L}_\beta(x, y)) \geq -\epsilon_s, \tag{13}$$

for $\epsilon_s \geq 0$. Naturally, for second-order stationarity, we use $\lambda_{\min}(\nabla_{xx} \mathcal{L}_\beta(x, y))$ as the stopping criterion.

**Smoothness lemma.** This next result controls the smoothness of $\mathcal{L}_\beta(\cdot, y)$ for a fixed $y$. The proof is standard but nevertheless is included in Appendix C for completeness.

**Lemma 2.1 (smoothness).** *For fixed $y \in \mathbb{R}^m$ and $\rho, \rho' \geq 0$, it holds that*

$$\|\nabla_x \mathcal{L}_\beta(x, y) - \nabla_x \mathcal{L}_\beta(x', y)\| \leq \lambda_\beta \|x - x'\|, \tag{14}$$

*for every $x, x' \in \{x'' : \|x''\| \leq \rho, \|A(x'')\| \leq \rho'\}$, where*

$$\lambda_\beta \leq \lambda_f + \sqrt{m}\lambda_A \|y\| + (\sqrt{m}\lambda_A \rho' + d\lambda_A'^2)\beta =: \lambda_f + \sqrt{m}\lambda_A \|y\| + \lambda''(A, \rho, \rho')\beta. \tag{15}$$

*Above, $\lambda_f, \lambda_A$ were defined in (4) and*

$$\lambda_A' := \max_{\|x\| \leq \rho} \|DA(x)\|. \tag{16}$$

# 3 Algorithm

To solve the equivalent formulation of (1) presented in (5), we propose the inexact ALM (iALM), detailed in Algorithm 1. At the $k^{\text{th}}$ iteration, Step 2 of Algorithm 1 calls a solver that finds an approximate stationary point of the augmented Lagrangian $\mathcal{L}_{\beta_k}(\cdot, y_k)$ with the accuracy of $\epsilon_{k+1}$, and this accuracy gradually increases in a controlled fashion. The increasing sequence of penalty weights $\{\beta_k\}_k$ and the dual update (Steps 4 and 5) are responsible for continuously enforcing the constraints in (1). The appropriate choice for $\{\beta_k\}_k$ will be specified in Corrollary Sections A.1 and A.2.

The particular choice of the dual step sizes $\{\sigma_k\}_k$ in Algorithm 1 ensures that the dual variable $y_k$ remains bounded.

---
**Algorithm 1** Inexact ALM

---
**Input:** Non-decreasing, positive, unbounded sequence $\{\beta_k\}_{k\geq 1}$, stopping thresholds $\tau_f, \tau_s > 0$.

**Initialization:** Primal variable $x_1 \in \mathbb{R}^d$, dual variable $y_0 \in \mathbb{R}^m$, dual step size $\sigma_1 > 0$.

**for** $k = 1, 2, \ldots$ **do**

    1.   **(Update tolerance)** $\epsilon_{k+1} = 1/\beta_k$.

    2.   **(Inexact primal solution)** Obtain $x_{k+1} \in \mathbb{R}^d$ such that

$$\text{dist}(-\nabla_x \mathcal{L}_{\beta_k}(x_{k+1}, y_k), \partial g(x_{k+1})) \leq \epsilon_{k+1}$$

    for first-order stationarity

$$\lambda_{\min}(\nabla_{xx} \mathcal{L}_{\beta_k}(x_{k+1}, y_k)) \geq -\epsilon_{k+1}$$

    for second-order-stationarity, if $g = 0$ in (1).

    3.   **(Update dual step size)**

$$\sigma_{k+1} = \sigma_1 \min\left(\frac{\|A(x_1)\| \log^2 2}{\|A(x_{k+1})\|(k+1)\log^2(k+2)}, 1\right).$$

    4.   **(Dual ascent)** $y_{k+1} = y_k + \sigma_{k+1} A(x_{k+1})$.

    5.   **(Stopping criterion)** If

$$\text{dist}(-\nabla_x \mathcal{L}_{\beta_k}(x_{k+1}), \partial g(x_{k+1})) + \|A(x_{k+1})\| \leq \tau_f,$$

    for first-order stationarity and if also $\lambda_{\min}(\nabla_{xx}\mathcal{L}_{\beta_k}(x_{k+1}, y_k)) \geq -\tau_s$ for second-order stationarity, then quit and return $x_{k+1}$ as an (approximate) stationary point of (5).

**end for**

---

# 4 Convergence Rate

This section presents the total iteration complexity of Algorithm 1 for finding first and second-order stationary points of problem (5). All the proofs are deferred to Appendix B. Theorem 4.1 characterizes the convergence rate of Algorithm 1 for finding stationary points in the number of outer iterations.

**Theorem 4.1. (convergence rate)** *For integers $2 \leq k_0 \leq k_1$, consider the interval $K = [k_0 : k_1]$, and let $\{x_k\}_{k\in K}$ be the output sequence of Algorithm 1 on the interval $K$.[2] Let also $\rho := \sup_{k\in[K]} \|x_k\|$.[3] Suppose that $f$ and $A$ satisfy (4) and let*

$$\lambda'_f = \max_{\|x\|\leq\rho} \|\nabla f(x)\|, \qquad \lambda'_A = \max_{\|x\|\leq\rho} \|DA(x)\|, \tag{17}$$

*be the (restricted) Lipschitz constants of $f$ and $A$, respectively. With $\nu > 0$, assume that*

$$\nu\|A(x_k)\| \leq \text{dist}\left(-DA(x_k)^\top A(x_k), \frac{\partial g(x_k)}{\beta_{k-1}}\right), \tag{18}$$

*for every $k \in K$. We consider two cases:*

- *If a first-order solver is used in Step 2, then $x_k$ is an $(\epsilon_{k,f}, \beta_k)$ first-order stationary point of (5) with*

$$\epsilon_{k,f} = \frac{1}{\beta_{k-1}} \left( \frac{2(\lambda'_f + \lambda'_A y_{\max})(1 + \lambda'_A \sigma_k)}{\nu} + 1 \right) =: \frac{Q(f, g, A, \sigma_1)}{\beta_{k-1}}, \qquad (19)$$

*for every $k \in K$, where $y_{\max}(x_1, y_0, \sigma_1) := \|y_0\| + c\|A(x_1)\|$.*

- *If a second-order solver is used in Step 2, then $x_k$ is an $(\epsilon_{k,f}, \epsilon_{k,s}, \beta_k)$ second-order stationary point of (5) with $\epsilon_{k,s}$ specified above and with*

$$\epsilon_{k,s} = \epsilon_{k-1} + \sigma_k \sqrt{m} \lambda_A \frac{2\lambda'_f + 2\lambda'_A y_{\max}}{\nu \beta_{k-1}} = \frac{\nu + \sigma_k \sqrt{m} \lambda_A 2\lambda'_f + 2\lambda'_A y_{\max}}{\nu \beta_{k-1}} =: \frac{Q'(f, g, A, \sigma_1)}{\beta_{k-1}}. \tag{20}$$

Theorem 4.1 states that Algorithm 1 converges to a (first- or second-) order stationary point of (5) at the rate of $1/\beta_k$, further specified in Corollary 4.2 and Corollary 4.3. A few remarks are in order about Theorem 4.1.

**Regularity.** The key geometric condition in Theorem 4.1 is (18) which, broadly speaking, ensures that the primal updates of Algorithm 1 reduce the feasibility gap as the penalty weight $\beta_k$ grows. We will verify this condition for several examples in Appendices D and E.

This condition in (18) is closely related to those in the existing literature. In the special case where $g = 0$ in (1), (18) reduces to;

$$\|DA(x)^\top A(x)\| \geq \nu \|A(x)\|. \tag{21}$$

*Polyak-Lojasiewicz (PL) condition [32].* Consider the problem with $\lambda_{\tilde{f}}$-smooth objective,

$$\min_{x \in \mathbb{R}^d} \tilde{f}(x).$$

$\tilde{f}(x)$ satisfies the PL inequality if the following holds for some $\mu > 0$,

$$\frac{1}{2}\|\nabla \tilde{f}(x)\|^2 \geq \mu(\tilde{f}(x) - \tilde{f}^*), \quad \forall x \qquad \text{(PL inequality)}$$

This inequality implies that gradient is growing faster than a quadratic as we move away from the optimal. Assuming that the feasible set $\{x : A(x) = 0\}$ is non-empty, it is easy to verify that 21 is equivalent to the PL condition for minimizing $\tilde{f}(x) = \frac{1}{2}\|A(x)\|^2$ with $\nu = \sqrt{2\mu}$ [32].

PL condition itself is a special case of Kurdyka-Lojasiewicz with $\theta = 1/2$, see [66, Definition 1.1]. When $g = 0$, it is also easy to see that (18) is weaker than the Mangasarian-Fromovitz (MF) condition in nonlinear optimization [10, Assumption 1]. Moreover, when $g$ is the indicator on a convex set, (18) is a consequence of the *basic constraint qualification* in [55], which itself generalizes the MF condition to the case when $g$ is an indicator function of a convex set.

We may think of (18) as a local condition, which should hold within a neighborhood of the constraint set $\{x : A(x) = 0\}$ rather than everywhere in $\mathbb{R}^d$. Indeed, the iteration count $k$ appears in (18) to reflect this local nature of the condition. Similar kind of arguments on the regularity condition also appear in [10]. There is also a constant complexity algorithm in [10] to reach so-called "information zone", which supplements Theorem 4.1.

**Penalty method.** A classical algorithm to solve (1) is the penalty method, which is characterized by the absence of the dual variable ($y = 0$) in (6). Indeed, ALM can be interpreted as an adaptive penalty or smoothing method with a variable center determined by the dual variable. It is worth noting that, with the same proof technique, one can establish the same convergence rate of Theorem 4.1 for the penalty method. However, while both methods have the same convergence rate in theory, we ignore the uncompetitive penalty method since it is significantly outperformed by iALM in practice.

**Computational complexity.** Theorem 4.1 specifies the number of (outer) iterations that Algorithm 1 requires to reach a near-stationary point of problem (6) with a prescribed precision and, in particular, specifies the number of calls made to the solver in Step 2. In this sense, Theorem 4.1 does

not fully capture the computational complexity of Algorithm 1, as it does not take into account the computational cost of the solver in Step 2.

To better understand the total iteration complexity of Algorithm 1, we consider two scenarios in the following. In the first scenario, we take the solver in Step 2 to be the Accelerated Proximal Gradient Method (APGM), a well-known first-order algorithm [27]. In the second scenario, we will use the second-order trust region method developed in [17]. We have the following two corollaries showing the total complexity of our algorithm to reach first and second-order stationary points. Appendix A contains the proofs and more detailed discussion for the complexity results.

**Corollary 4.2** (First-order optimality). *For $b > 1$, let $\beta_k = b^k$ for every $k$. If we use APGM from [27] for Step 2 of Algorithm 1, the algorithm finds an $(\epsilon_f, \beta_k)$ first-order stationary point of* (5)*, after $T$ calls to the first-order oracle, where*

$$T = \mathcal{O}\left( \frac{Q^3 \rho^2}{\epsilon^3} \log_b \left( \frac{Q}{\epsilon} \right) \right) = \tilde{\mathcal{O}}\left( \frac{Q^3 \rho^2}{\epsilon^3} \right). \tag{22}$$

For Algorithm 1 to reach a near-stationary point with an accuracy of $\epsilon_f$ in the sense of (9) and with the lowest computational cost, we therefore need to perform only one iteration of Algorithm 1, with $\beta_1$ specified as a function of $\epsilon_f$ by (19) in Theorem 4.1. In general, however, the constants in (19) are unknown and this approach is thus not feasible. Instead, the homotopy approach taken by Algorithm 1 ensures achieving the desired accuracy by gradually increasing the penalty weight. This homotopy approach increases the computational cost of Algorithm 1 only by a factor logarithmic in the $\epsilon_f$, as detailed in the proof of Corollary 4.2.

**Corollary 4.3** (Second-order optimality). *For $b > 1$, let $\beta_k = b^k$ for every $k$. We assume that*

$$\mathcal{L}_\beta(x_1, y) - \min_x \mathcal{L}_\beta(x, y) \le L_u, \qquad \forall \beta. \tag{23}$$

*If we use the trust region method from [17] for Step 2 of Algorithm 1, the algorithm finds an $\epsilon$-second-order stationary point of* (5) *in $T$ calls to the second-order oracle where*

$$T = \mathcal{O}\left( \frac{L_u Q'^5}{\epsilon^5} \log_b \left( \frac{Q'}{\epsilon} \right) \right) = \widetilde{\mathcal{O}}\left( \frac{L_u Q'^5}{\epsilon^5} \right). \tag{24}$$

**Remark.** These complexity results for first and second-order are stationarity with respect to (6). We note that these complexities match [18] and [7]. However, the stationarity criteria and the definition of dual variable in these papers differ from ours. We include more discussion on this in the Appendix.

**Effect of $\beta_k$ in 18.** We consider two cases, when $g$ is the indicator of a convex set (or 0), the subdifferential set will be a cone (or 0), thus $\beta_k$ will not have an effect. On the other hand, when $g$ is a convex and Lipschitz contiunous function defined on the whole space, subdifferential set will be bounded [54, Theorem 23.4]. This will introduce an error term in 18 that is of the order $(1/\beta_k)$. One can see that $b^k$ choice for $\beta_k$ causes a linear decrease in this error term. In fact, all the examples in this paper fall into the first case.

# 5 Related Work

ALM has a long history in the optimization literature, dating back to [29, 51]. In the special case of (1) with a convex function $f$ and a linear operator $A$, standard, inexact, and linearized versions of ALM have been extensively studied [36, 41, 61, 65].

Classical works on ALM focused on the general template of (1) with nonconvex $f$ and nonlinear $A$, with arguably stronger assumptions and required exact solutions to the subproblems of the form (7), which appear in Step 2 of Algorithm 1, see for instance [4].

A similar analysis was conducted in [22] for the general template of (1). The authors considered inexact ALM and proved convergence rates for the outer iterates, under specific assumptions on the initialization of the dual variable. However, in contrast, the authors did not analyze how to solve the subproblems inexactly and did not provide total complexity results with verifiable conditions.

Problem (1) with similar assumptions to us is also studied in [7] and [18] for first-order and second-order stationarity, respectively, with explicit iteration complexity analysis. As we have mentioned

in Section 4, our iteration complexity results matches these theoretical algorithms with a simpler algorithm and a simpler analysis. In addition, these algorithms require setting final accuracies since they utilize this information in the algorithm while our Algorithm 1 does not set accuracies a priori.

[16] also considers the same template (1) for first-order stationarity with a penalty-type method instead of ALM. Even though the authors show $\mathcal{O}(1/\epsilon^2)$ complexity, this result is obtained by assuming that the penalty parameter remains bounded. We note that such an assumption can also be used to improve our complexity results to match theirs.

[10] studies the general template (1) with specific assumptions involving local error bound conditions for the (1). These conditions are studied in detail in [9], but their validity for general SDPs (2) has never been established. This work also lacks the total iteration complexity analysis presented here.

Another work [20] focused on solving (1) by adapting the primal-dual method of Chambolle and Pock [19]. The authors proved the convergence of the method and provided convergence rate by imposing error bound conditions on the objective function that do not hold for standard SDPs.

[14, 15] is the first work that proposes the splitting $X = UU^\top$ for solving SDPs of the form (2). Following these works, the literature on Burer-Monteiro (BM) splitting for the large part focused on using ALM for solving the reformulated problem (3).

However, this proposal has a few drawbacks: First, it requires exact solutions in Step 2 of Algorithm 1 in theory, which in practice is replaced with inexact solutions. Second, their results only establish convergence without providing the rates. In this sense, our work provides a theoretical understanding of the BM splitting with inexact solutions to Step 2 of Algorithm 1 and complete iteration complexities.

[6, 48] are among the earliest efforts to show convergence rates for BM splitting, focusing on the special case of SDPs without any linear constraints. For these specific problems, they prove the convergence of gradient descent to global optima with convergence rates, assuming favorable initialization. These results, however, do not apply to general SDPs of the form (2) where the difficulty arises due to the linear constraints.

Another popular method for solving SDPs are due to [12, 11, 13], focusing on the case where the constraints in (1) can be written as a Riemannian manifold after BM splitting. In this case, the authors apply the Riemannian gradient descent and Riemannian trust region methods for obtaining first- and second-order stationary points, respectively. They obtain $\mathcal{O}(1/\epsilon^2)$ complexity for finding first-order stationary points and $\mathcal{O}(1/\epsilon^3)$ complexity for finding second-order stationary points.

While these complexities appear better than ours, the smooth manifold requirement in these works is indeed restrictive. In particular, this requirement holds for max-cut and generalized eigenvalue problems, but it is not satisfied for other important SDPs such as quadratic programming (QAP), optimal power flow and clustering with general affine constraints. In addition, as noted in [11], per iteration cost of their method for max-cut problem is an astronomical $\mathcal{O}(d^6)$.

Lastly, there also exists a line of work for solving SDPs in their original convex formulation, in a storage efficient way [42, 68, 69]. These works have global optimality guarantees by their virtue of directly solving the convex formulation. On the downside, these works require the use of eigenvalue routines and exhibit significantly slower convergence as compared to nonconvex approaches [31].

# 6  Numerical Evidence

We first begin with a caveat: It is known that quasi-Newton methods, such as BFGS and lBFGS, might not converge for nonconvex problems [21, 38]. For this reason, we have used the trust region method as the second-order solver in our analysis in Section 4, which is well-studied for nonconvex problems [17]. Empirically, however, BFGS and lBGFS are extremely successful and we have therefore opted for those solvers in this section since the subroutine does not affect Theorem 4.1 as long as the subsolver performs well in practice.

## 6.1  Clustering

Given data points $\{z_i\}_{i=1}^n$, the entries of the corresponding Euclidean distance matrix $D \in \mathbb{R}^{n \times n}$ are $D_{i,j} = \|z_i - z_j\|^2$. Clustering is then the problem of finding a co-association matrix $Y \in \mathbb{R}^{n \times n}$ such that $Y_{ij} = 1$ if points $z_i$ and $z_j$ are within the same cluster and $Y_{ij} = 0$ otherwise. In [50], the

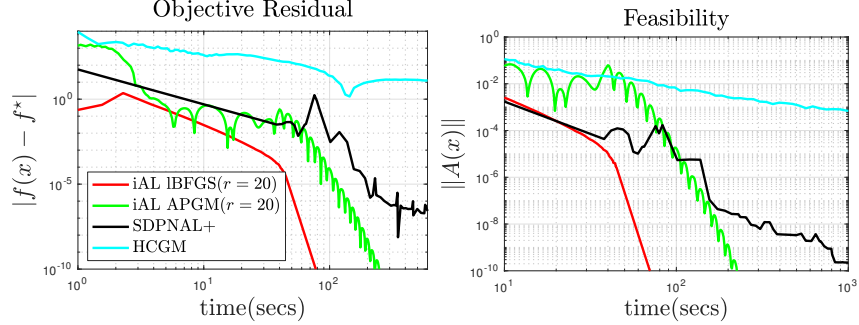

Figure 1: Clustering running time comparison.

authors provide a SDP relaxation of the clustering problem, specified as

$$\min_{Y \in \mathbb{R}^{nxn}} \text{tr}(DY) \quad \text{s.t.} \quad Y\mathbf{1} = \mathbf{1}, \ \text{tr}(Y) = s, \ Y \succeq 0, \ Y \geq 0, \tag{25}$$

where $s$ is the number of clusters and $Y$ is both positive semidefinite and has nonnegative entries. Standard SDP solvers do not scale well with the number of data points $n$, since they often require projection onto the semidefinite cone with the complexity of $\mathcal{O}(n^3)$. We instead use the BM factorization to solve (25), sacrificing convexity to reduce the computational complexity. More specifically, we solve the program

$$\min_{V \in \mathbb{R}^{n \times r}} \text{tr}(DVV^\top) \quad \text{s.t.} \quad VV^\top \mathbf{1} = \mathbf{1}, \ \|V\|_F^2 \leq s, \ V \geq 0, \tag{26}$$

where $\mathbf{1} \in \mathbb{R}^n$ is the vector of all ones. Note that $Y \geq 0$ in (25) is replaced above by the much stronger but easier-to-enforce constraint $V \geq 0$ in (26), see [35] for the reasoning behind this relaxation. Now, we can cast (26) as an instance of (1). Indeed, for every $i \leq n$, let $x_i \in \mathbb{R}^r$ denote the $i$th row of $V$. We next form $x \in \mathbb{R}^d$ with $d = nr$ by expanding the factorized variable $V$, namely, $x := [x_1^\top, \cdots, x_n^\top]^\top \in \mathbb{R}^d$, and then set

$$f(x) = \sum_{i,j=1}^n D_{i,j} \langle x_i, x_j \rangle, \qquad g = \delta_C, \qquad A(x) = [x_1^\top \sum_{j=1}^n x_j - 1, \cdots, x_n^\top \sum_{j=1}^n x_j - 1]^\top,$$

where $C$ is the intersection of the positive orthant in $\mathbb{R}^d$ with the Euclidean ball of radius $\sqrt{s}$. In Appendix D, we verify that Theorem 4.1 applies to (1) with $f, g, A$ specified above.

In our simulations, we use two different solvers for Step 2 of Algorithm 1, namely, APGM and lBFGS. APGM is a solver for nonconvex problems of the form (7) with convergence guarantees to first-order stationarity, as discussed in Section 4. lBFGS is a limited-memory version of BFGS algorithm in [24] that approximately leverages the second-order information of the problem. We compare our approach against the following convex methods:

- HCGM: Homotopy-based Conditional Gradient Method in [69] which directly solves (25).
- SDPNAL+: A second-order augmented Lagrangian method for solving SDP's with nonnegativity constraints [67].

As for the dataset, our experimental setup is similar to that described by [39]. We use the publicly-available fashion-MNIST data in [64], which is released as a possible replacement for the MNIST handwritten digits. Each data point is a $28 \times 28$ gray-scale image, associated with a label from ten classes, labeled from 0 to 9. First, we extract the meaningful features from this dataset using a simple two-layer neural network with a sigmoid activation function. Then, we apply this neural network to 1000 test samples from the same dataset, which gives us a vector of length 10 for each data point, where each entry represents the posterior probability for each class. Then, we form the $\ell_2$ distance matrix $D$ from these probability vectors. The solution rank for the template (25) is known and it is equal to number of clusters $k$ [35, Theorem 1]. As discussed in [60], setting rank $r > k$ leads more accurate reconstruction in expense of speed. Therefore, we set the rank to 20. For iAL lBFGS, we used $\beta_1 = 1$ and $\sigma_1 = 10$ as the initial penalty weight and dual step size, respectively. For HCGM,

we used $\beta_0 = 1$ as the initial smoothness parameter. We have run SDPNAL+ solver with $10^{-12}$ tolerance. The results are depicted in Figure 1. We implemented 3 algorithms on MATLAB and used the software package for SDPNAL+ which contains mex files. It is predictable that the performance of our nonconvex approach would even improve by using mex files.

## 6.2 Additional demonstrations

We provide several additional experiments in Appendix E. Section E.1 discusses a novel nonconvex relaxation of the standard basis pursuit template which performs comparable to the state of the art convex solvers. In Section E.2, we provide fast numerical solutions to the generalized eigenvalue problem. In Section E.3, we give a contemporary application example that our template applies, namely, denoising with generative adversarial networks. Finally, we provide improved bounds for sparse quadratic assignment problem instances in Section E.4.

# 7   Conclusions

In this work, we have proposed and analyzed an inexact augmented Lagrangian method for solving nonconvex optimization problems with nonlinear constraints. We prove convergence to the first and second order stationary points of the augmented Lagrangian function, with explicit complexity estimates. Even though the relation of stationary points and global optima is not well-understood in the literature, we find out that the algorithm has fast convergence behavior to either global minima or local minima in a wide variety of numerical experiments.

# Acknowledgements

The authors would like to thank Nicolas Boumal and Nadav Hallak for the helpful suggestions.

This project has received funding from the European Research Council (ERC) under the European Union's Horizon 2020 research and innovation programme (grant agreement n° 725594 - time-data) and was supported by the Swiss National Science Foundation (SNSF) under grant number 200021_178865/1. This project was also sponsored by the Department of the Navy, Office of Naval Research (ONR) under a grant number N62909-17-1-2111 and was supported by Hasler Foundation Program: Cyber Human Systems (project number 16066). This research was supported by the PhD fellowship program of the Swiss Data Science Center (SDSC) under grant lD number P18-07.

## Footnotes

[1]BFGS is in fact a quasi-Newton method that emulates second-order information.

[2]The choice of $k_1 = \infty$ is valid here too.

[3]If necessary, to ensure that $\rho < \infty$, one can add a small factor of $\|x\|^2$ to $\mathcal{L}_\beta$ in (6). Then it is easy to verify that the iterates of Algorithm 1 remain bounded, provided that the initial penalty weight $\beta_0$ is large enough, $\sup_x \|\nabla f(x)\|/\|x\| < \infty$, $\sup_x \|A(x)\| < \infty$, and $\sup_x \|DA(x)\| < \infty$.

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
