[Supplementary Material]

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

# A Complexity Results

## A.1 First-Order Optimality

Let us first consider the case where the solver in Step 2 is is the first-order algorithm APGM, described in detail in [27]. At a high level, APGM makes use of $\nabla_x \mathcal{L}_\beta(x, y)$ in (6), the proximal operator $\text{prox}_g$, and the classical Nesterov acceleration [43] to reach first-order stationarity for the subproblem in (7). Suppose that $g = \delta_{\mathcal{X}}$ is the indicator function on a bounded convex set $\mathcal{X} \subset \mathbb{R}^d$ and let

$$\rho = \max_{x \in \mathcal{X}} \|x\|, \tag{27}$$

be the radius of a ball centered at the origin that includes $\mathcal{X}$. Then, adapting the results in [27] to our setup, APGM reaches $x_k$ in Step 2 of Algorithm 1 after

$$\mathcal{O}\left(\frac{\lambda_{\beta_k}^2 \rho^2}{\epsilon_{k+1}}\right) \tag{28}$$

(inner) iterations, where $\lambda_{\beta_k}$ denotes the Lipschitz constant of $\nabla_x \mathcal{L}_{\beta_k}(x, y)$, bounded in (15). For the clarity of the presentation, we have used a looser bound in (28) compared to [27]. Using (28), we derive the following corollary, describing the total iteration complexity of Algorithm 1 in terms of the number calls made to the first-order oracle in APGM.

**Corollary A.1.** *For $b > 1$, let $\beta_k = b^k$ for every $k$. If we use APGM from [27] for Step 2 of Algorithm 1, the algorithm finds an $(\epsilon_f, \beta_k)$ first-order stationary point, after $T$ calls to the first-order oracle, where*

$$T = \mathcal{O}\left(\frac{Q^3 \rho^2}{\epsilon^3} \log_b\left(\frac{Q}{\epsilon}\right)\right) = \tilde{\mathcal{O}}\left(\frac{Q^3 \rho^2}{\epsilon^3}\right). \tag{29}$$

*Proof.* Let $K$ denote the number of (outer) iterations of Algorithm 1 and let $\epsilon_f$ denote the desired accuracy of Algorithm 1, see (9). Recalling Theorem 4.1, we can then write that

$$\epsilon_f = \frac{Q}{\beta_K}, \tag{30}$$

or, equivalently, $\beta_K = Q/\epsilon_f$. We now count the number of total (inner) iterations $T$ of Algorithm 1 to reach the accuracy $\epsilon_f$. From (15) and for sufficiently large $k$, recall that $\lambda_{\beta_k} \leq \lambda'' \beta_k$ is the smoothness parameter of the augmented Lagrangian. Then, from (28) ad by summing over the outer iterations, we bound the total number of (inner) iterations of Algorithm 1 as

$$
\begin{aligned}
T &= \sum_{k=1}^{K} \mathcal{O}\left(\frac{\lambda_{\beta_{k-1}}^2 \rho^2}{\epsilon_k}\right) \\
&= \sum_{k=1}^{K} \mathcal{O}\left(\beta_{k-1}^3 \rho^2\right) \qquad \text{(Step 1 of Algorithm 1)} \\
&\leq \mathcal{O}\left(K \beta_{K-1}^3 \rho^2\right) \qquad (\{\beta_k\}_k \text{ is increasing}) \\
&\leq \mathcal{O}\left(\frac{K Q^3 \rho^2}{\epsilon_f^3}\right). \qquad \text{(see (30))}
\end{aligned} \tag{31}
$$

In addition, if we specify $\beta_k = b^k$ for all $k$, we can further refine $T$. Indeed,

$$\beta_K = b^K \implies K = \log_b\left(\frac{Q}{\epsilon_f}\right), \tag{32}$$

which, after substituting into (31) gives the final bound in Corollary 4.2. □

## A.2 Second-Order Optimality

Let us now consider the second-order optimality case where the solver in Step 2 is the the trust region method developed in [17]. Trust region method minimizes a quadratic approximation of the function

within a dynamically updated trust-region radius. Second-order trust region method that we consider in this section makes use of Hessian (or an approximation of Hessian) of the augmented Lagrangian in addition to first order oracles.

As shown in [45], finding approximate second-order stationary points of convex-constrained problems is in general NP-hard. For this reason, we focus in this section on the special case of (1) with $g = 0$.

Let us compute the total computational complexity of Algorithm 1 with the trust region method in Step 2, in terms of the number of calls made to the second-order oracle. By adapting the result in [17] to our setup, we find that the number of (inner) iterations required in Step 2 of Algorithm 1 to produce $x_{k+1}$ is

$$\mathcal{O}\left( \frac{\lambda_{\beta_k,H}^2 (\mathcal{L}_{\beta_k}(x_1, y) - \min_x \mathcal{L}_{\beta_k}(x, y))}{\epsilon_k^3} \right), \tag{33}$$

where $\lambda_{\beta,H}$ is the Lipschitz constant of the Hessian of the augmented Lagrangian, which is of the order of $\beta$, as can be proven similar to Lemma 2.1 and $x_1$ is the initial iterate of the given outer loop. In [17], the term $\mathcal{L}_\beta(x_1, y) - \min_x \mathcal{L}_\beta(x, y)$ is bounded by a constant independent of $\epsilon$. We assume a uniform bound for this quantity for every $\beta_k$, instead of for one value of $\beta_k$ as in [17]. Using (33) and Theorem 4.1, we arrive at the following:

**Corollary A.2.** *For $b > 1$, let $\beta_k = b^k$ for every $k$. We assume that*

$$\mathcal{L}_\beta(x_1, y) - \min_x \mathcal{L}_\beta(x, y) \le L_u, \qquad \forall \beta. \tag{34}$$

*If we use the trust region method from [17] for Step 2 of Algorithm 1, the algorithm finds an $\epsilon$-second-order stationary point of (1) in $T$ calls to the second-order oracle where*

$$T = \mathcal{O}\left( \frac{L_u Q'^5}{\epsilon^5} \log_b \left( \frac{Q'}{\epsilon} \right) \right) = \widetilde{\mathcal{O}}\left( \frac{L_u Q'^5}{\epsilon^5} \right). \tag{35}$$

Before closing this section, we note that the remark after Corollary 4.2 applies here as well.

## A.3 Approximate optimality of (1).

Corollary 4.2 establishes the iteration complexity of Algorithm 1 to reach approximate first-order stationarity for the equivalent formulation of (1) presented in (5). Unlike the exact case, approximate first-order stationarity in (5) does not immediately lend itself to approximate stationarity in (1), and the study of approximate stationarity for the penalized problem (special case of our setting with dual variable set to 0) has also precedent in [5]. For a precedent in convex optimization for relating the convergence in augmented Lagrangian to the constrained problem using duality, see [62]. For the second-order case, it is in general not possible to establish approximate second-order optimality for (5) from Corollary 4.3, with the exception of linear constraints. [45] provides an hardness result by showing that checking an approximate second-order stationarity is NP-hard.

## B Proof of Theorem 4.1

For every $k \ge 2$, recall from (6) and Step 2 of Algorithm 1 that $x_k$ satisfies

$$\text{dist}(-\nabla f(x_k) - DA(x_k)^\top y_{k-1}$$
$$- \beta_{k-1} DA(x_k)^\top A(x_k), \partial g(x_k))$$
$$= \text{dist}(-\nabla_x \mathcal{L}_{\beta_{k-1}}(x_k, y_{k-1}), \partial g(x_k)) \le \epsilon_k. \tag{36}$$

With an application of the triangle inequality, it follows that

$$\text{dist}(-\beta_{k-1} DA(x_k)^\top A(x_k), \partial g(x_k))$$
$$\le \|\nabla f(x_k)\| + \|DA(x_k)^\top y_{k-1}\| + \epsilon_k, \tag{37}$$

which in turn implies that

$$\text{dist}(-DA(x_k)^\top A(x_k), \partial g(x_k)/\beta_{k-1})$$
$$\le \frac{\|\nabla f(x_k)\|}{\beta_{k-1}} + \frac{\|DA(x_k)^\top y_{k-1}\|}{\beta_{k-1}} + \frac{\epsilon_k}{\beta_{k-1}}$$
$$\le \frac{\lambda_f' + \lambda_A' \|y_{k-1}\| + \epsilon_k}{\beta_{k-1}}, \tag{38}$$

where $\lambda'_f$, $\lambda'_A$ were defined in (17). We next translate (38) into a bound on the feasibility gap $\|A(x_k)\|$. Using the regularity condition (18), the left-hand side of (38) can be bounded below as

$$\text{dist}(-DA(x_k)^\top A(x_k), \partial g(x_k)/\beta_{k-1}) \geq \nu\|A(x_k)\|. \qquad \text{(see (18))} \tag{39}$$

By substituting (39) back into (38), we find that

$$\|A(x_k)\| \leq \frac{\lambda'_f + \lambda'_A\|y_{k-1}\| + \epsilon_k}{\nu\beta_{k-1}}. \tag{40}$$

In words, the feasibility gap is directly controlled by the dual sequence $\{y_k\}_k$. We next establish that the dual sequence is bounded. Indeed, for every $k \in K$, note that

$$\|y_k\| = \|y_0 + \sum_{i=1}^{k} \sigma_i A(x_i)\| \quad \text{(Step 5 of Algorithm 1)}$$

$$\leq \|y_0\| + \sum_{i=1}^{k} \sigma_i\|A(x_i)\| \qquad \text{(triangle inequality)}$$

$$\leq \|y_0\| + \sum_{i=1}^{k} \frac{\|A(x_1)\|\log^2 2}{k\log^2(k+1)} \quad \text{(Step 4)}$$

$$\leq \|y_0\| + c\|A(x_1)\|\log^2 2 =: y_{\max}, \tag{41}$$

where

$$c \geq \sum_{i=1}^{\infty} \frac{1}{k\log^2(k+1)}. \tag{42}$$

Substituting (41) back into (40), we reach

$$\|A(x_k)\| \leq \frac{\lambda'_f + \lambda'_A y_{\max} + \epsilon_k}{\nu\beta_{k-1}}$$

$$\leq \frac{2\lambda'_f + 2\lambda'_A y_{\max}}{\nu\beta_{k-1}}, \tag{43}$$

where the second line above holds if $k_0$ is large enough, which would in turn guarantees that $\epsilon_k = 1/\beta_{k-1}$ is sufficiently small since $\{\beta_k\}_k$ is increasing and unbounded. It remains to control the first term in (10). To that end, after recalling Step 2 of Algorithm 1 and applying the triangle inequality, we can write that

$$\text{dist}(-\nabla_x \mathcal{L}_{\beta_{k-1}}(x_k, y_k), \partial g(x_k))$$
$$\leq \text{dist}(-\nabla_x \mathcal{L}_{\beta_{k-1}}(x_k, y_{k-1}), \partial g(x_k))$$
$$+ \|\nabla_x \mathcal{L}_{\beta_{k-1}}(x_k, y_k) - \nabla_x \mathcal{L}_{\beta_{k-1}}(x_k, y_{k-1})\|. \tag{44}$$

The first term on the right-hand side above is bounded by $\epsilon_k$, by Step 5 of Algorithm 1. For the second term on the right-hand side of (44), we write that

$$\|\nabla_x \mathcal{L}_{\beta_{k-1}}(x_k, y_k) - \nabla_x \mathcal{L}_{\beta_{k-1}}(x_k, y_{k-1})\|$$
$$= \|DA(x_k)^\top(y_k - y_{k-1})\| \qquad \text{(see (6))}$$
$$\leq \lambda'_A\|y_k - y_{k-1}\| \qquad \text{(see (17))}$$
$$= \lambda'_A \sigma_k\|A(x_k)\| \qquad \text{(see Step 5 of Algorithm 1)}$$
$$\leq \frac{2\lambda'_A \sigma_k}{\nu\beta_{k-1}}(\lambda'_f + \lambda'_A y_{\max}). \qquad \text{(see (43))} \tag{45}$$

By combining (44,45), we find that

$$\text{dist}(\nabla_x \mathcal{L}_{\beta_{k-1}}(x_k, y_k), \partial g(x_k))$$
$$\leq \frac{2\lambda'_A \sigma_k}{\nu\beta_{k-1}}(\lambda'_f + \lambda'_A y_{\max}) + \epsilon_k. \tag{46}$$

By combining (43,46), we find that

$$\text{dist}(-\nabla_x \mathcal{L}_{\beta_{k-1}}(x_k, y_k), \partial g(x_k)) + \|A(x_k)\|$$

$$\leq \left( \frac{2\lambda'_A \sigma_k}{\nu \beta_{k-1}} (\lambda'_f + \lambda'_A y_{\max}) + \epsilon_k \right)$$

$$+ 2\left( \frac{\lambda'_f + \lambda'_A y_{\max}}{\nu \beta_{k-1}} \right). \tag{47}$$

Applying $\sigma_k \leq \sigma_1$, we find that

$$\text{dist}(-\nabla_x \mathcal{L}_{\beta_{k-1}}(x_k, y_k), \partial g(x_k)) + \|A(x_k)\|$$

$$\leq \frac{2\lambda'_A \sigma_1 + 2}{\nu \beta_{k-1}} (\lambda'_f + \lambda'_A y_{\max}) + \epsilon_k. \tag{48}$$

For the second part of the theorem, we use the Weyl's inequality and Step 5 of Algorithm 1 to write

$$\lambda_{\min}(\nabla_{xx}\mathcal{L}_{\beta_{k-1}}(x_k, y_{k-1})) \geq \lambda_{\min}(\nabla_{xx}\mathcal{L}_{\beta_{k-1}}(x_k, y_k))$$

$$- \sigma_k \| \sum_{i=1}^{m} A_i(x_k)\nabla^2 A_i(x_k) \|. \tag{49}$$

The first term on the right-hand side is lower bounded by $-\epsilon_{k-1}$ by Step 2 of Algorithm 1. We next bound the second term on the right-hand side above as

$$\sigma_k \| \sum_{i=1}^{m} A_i(x_k)\nabla^2 A_i(x_k) \|$$

$$\leq \sigma_k \sqrt{m} \max_i \|A_i(x_k)\| \|\nabla^2 A_i(x_k)\|$$

$$\leq \sigma_k \sqrt{m} \lambda_A \frac{2\lambda'_f + 2\lambda'_A y_{\max}}{\nu \beta_{k-1}},$$

where the last inequality is due to (4,43). Plugging into (49) gives

$$\lambda_{\min}(\nabla_{xx}\mathcal{L}_{\beta_{k-1}}(x_k, y_{k-1}))$$

$$\geq -\epsilon_{k-1} - \sigma_k \sqrt{m} \lambda_A \frac{2\lambda'_f + 2\lambda'_A y_{\max}}{\nu \beta_{k-1}},$$

which completes the proof of Theorem 4.1.

## C  Proof of Lemma 2.1

*Proof.* Note that

$$\mathcal{L}_\beta(x, y) = f(x) + \sum_{i=1}^{m} y_i A_i(x) + \frac{\beta}{2} \sum_{i=1}^{m} (A_i(x))^2, \tag{50}$$

which implies that

$$\nabla_x \mathcal{L}_\beta(x, y)$$

$$= \nabla f(x) + \sum_{i=1}^{m} y_i \nabla A_i(x) + \frac{\beta}{2} \sum_{i=1}^{m} A_i(x) \nabla A_i(x)$$

$$= \nabla f(x) + DA(x)^\top y + \beta DA(x)^\top A(x), \tag{51}$$

where $DA(x)$ is the Jacobian of $A$ at $x$. By taking another derivative with respect to $x$, we reach

$$\nabla_x^2 \mathcal{L}_\beta(x, y) = \nabla^2 f(x) + \sum_{i=1}^{m} (y_i + \beta A_i(x)) \nabla^2 A_i(x)$$

$$+ \beta \sum_{i=1}^{m} \nabla A_i(x) \nabla A_i(x)^\top. \tag{52}$$

It follows that

$$
\begin{aligned}
\|\nabla_x^2 \mathcal{L}_\beta(x,y)\| \\
\leq \|\nabla^2 f(x)\| + \max_i \|\nabla^2 A_i(x)\| \left(\|y\|_1 + \beta\|A(x)\|_1\right) \\
+ \beta \sum_{i=1}^{m} \|\nabla A_i(x)\|^2 \\
\leq \lambda_h + \sqrt{m}\lambda_A \left(\|y\| + \beta\|A(x)\|\right) + \beta\|DA(x)\|_F^2.
\end{aligned}
\tag{53}
$$

For every $x$ such that $\|x\| \leq \rho$ and $\|A(x)\| \leq \rho$, we conclude that

$$
\|\nabla_x^2 \mathcal{L}_\beta(x,y)\| \leq \lambda_f + \sqrt{m}\lambda_A \left(\|y\| + \beta\rho'\right) + \beta \max_{\|x\| \leq \rho} \|DA(x)\|_F^2,
\tag{54}
$$

which completes the proof of Lemma 2.1. $\qquad\square$

## D   Clustering

We only verify the condition in (18) here. Note that

$$
A(x) = VV^\top \mathbf{1} - \mathbf{1},
\tag{55}
$$

$$
\begin{aligned}
DA(x) &= \begin{bmatrix} w_{1,1}x_1^\top & \cdots & w_{1,n}x_1^\top \\ \vdots & & \\ w_{n,1}x_n^\top & \cdots & w_{n,n}1x_n^\top \end{bmatrix} \\
&= \begin{bmatrix} V & \cdots & V \end{bmatrix} + \begin{bmatrix} x_1^\top & & \\ & \ddots & \\ & & x_n^\top \end{bmatrix},
\end{aligned}
\tag{56}
$$

where $w_{i,i} = 2$ and $w_{i,j} = 1$ for $i \neq j$. In the last line above, $n$ copies of $V$ appear and the last matrix above is block-diagonal. For $x_k$, define $V_k$ accordingly and let $x_{k,i}$ be the $i$th row of $V_k$. Consequently,

$$
\begin{aligned}
DA(x_k)^\top A(x_k) &= \begin{bmatrix} (V_k^\top V_k - I_n)V_k^\top \mathbf{1} \\ \vdots \\ (V_k^\top V_k - I_n)V_k^\top \mathbf{1} \end{bmatrix} \\
&+ \begin{bmatrix} x_{k,1}(V_k V_k^\top \mathbf{1} - \mathbf{1})_1 \\ \vdots \\ x_{k,n}(V_k V_k^\top \mathbf{1} - \mathbf{1})_n \end{bmatrix},
\end{aligned}
\tag{57}
$$

where $I_n \in \mathbb{R}^{n \times n}$ is the identity matrix. Let us make a number of simplifying assumptions. First, we assume that $\|x_k\| < \sqrt{s}$ (which can be enforced in the iterates by replacing $C$ with $(1 - \epsilon)C$ for a small positive $\epsilon$ in the subproblems). Under this assumption, it follows that

$$
(\partial g(x_k))_i = \begin{cases} 0 & (x_k)_i > 0 \\ \{a : a \leq 0\} & (x_k)_i = 0, \end{cases} \quad i \leq d.
\tag{58}
$$

Second, we assume that $V_k$ has nearly orthonormal columns, namely, $V_k^\top V_k \approx I_n$. This can also be enforced in each iterate of Algorithm 1 and naturally corresponds to well-separated clusters. While a more fine-tuned argument can remove these assumptions, they will help us simplify the presentation

here. Under these assumptions, the (squared) right-hand side of (18) becomes

$$
\text{dist}\left(-DA(x_k)^\top A(x_k), \frac{\partial g(x_k)}{\beta_{k-1}}\right)^2
$$

$$
= \left\|\left(-DA(x_k)^\top A(x_k)\right)_+\right\|^2 \qquad (a_+ = \max(a,0))
$$

$$
= \left\|\begin{bmatrix} x_{k,1}(V_kV_k^\top\mathbf{1} - \mathbf{1})_1 \\ \vdots \\ x_{k,n}(V_kV_k^\top\mathbf{1} - \mathbf{1})_n \end{bmatrix}\right\|^2 \qquad (x_k \in C \Rightarrow x_k \geq 0)
$$

$$
= \sum_{i=1}^{n} \|x_{k,i}\|^2 (V_kV_k^\top\mathbf{1} - \mathbf{1})_i^2
$$

$$
\geq \min_i \|x_{k,i}\|^2 \cdot \sum_{i=1}^{n} (V_kV_k^\top\mathbf{1} - \mathbf{1})_i^2
$$

$$
= \min_i \|x_{k,i}\|^2 \cdot \|V_kV_k^\top\mathbf{1} - \mathbf{1}\|^2. \tag{59}
$$

Therefore, given a prescribed $\nu$, ensuring $\min_i \|x_{k,i}\| \geq \nu$ guarantees (18). When the algorithm is initialized close enough to the constraint set, there is indeed no need to separately enforce (59). In practice, often $n$ exceeds the number of true clusters and a more intricate analysis is required to establish (18) by restricting the argument to a particular subspace of $\mathbb{R}^n$.

## E  Additional Experiments

### E.1  Basis Pursuit

Basis Pursuit (BP) finds sparsest solutions of an under-determined system of linear equations by solving

$$
\min_z \|z\|_1 \quad \text{s.t.} \quad Bz = b, \tag{60}
$$

where $B \in \mathbb{R}^{n \times d}$ and $b \in \mathbb{R}^n$. Various primal-dual convex optimization algorithms are available in the literature to solve BP, including [61, 19]. We compare our algorithm against state-of-the-art primal-dual convex methods for solving (60), namely, Chambole-Pock [19], ASGARD [62] and ASGARD-DL [61].

Here, we take a different approach and cast (60) as an instance of (1). Note that any $z \in \mathbb{R}^d$ can be decomposed as $z = z^+ - z^-$, where $z^+, z^- \in \mathbb{R}^d$ are the positive and negative parts of $z$, respectively. Then consider the change of variables $z^+ = u_1^{\circ 2}$ and $z^- = u_2^{\circ 2} \in \mathbb{R}^d$, where $\circ$ denotes element-wise power. Next, we concatenate $u_1$ and $u_2$ as $x := [u_1^\top, u_2^\top]^\top \in \mathbb{R}^{2d}$ and define $\overline{B} := [B, -B] \in \mathbb{R}^{n \times 2d}$. Then, (60) is equivalent to (1) with

$$
f(x) = \|x\|^2, \quad g(x) = 0, \quad \text{s.t.} \quad A(x) = \overline{B}x^{\circ 2} - b. \tag{61}
$$

We draw the entries of $B$ independently from a zero-mean and unit-variance Gaussian distribution. For a fixed sparsity level $k$, the support of $z_* \in \mathbb{R}^d$ and its nonzero amplitudes are also drawn from the standard Gaussian distribution. Then the measurement vector is created as $b = Bz + \epsilon$, where $\epsilon$ is the noise vector with entries drawn independently from the zero-mean Gaussian distribution with variance $\sigma^2 = 10^{-6}$.

The results are compiled in Figure 2. Clearly, the performance of Algorithm 1 with a second-order solver for BP is comparable to the rest. It is, indeed, interesting to see that these type of nonconvex relaxations gives the solution of convex one and first order methods succeed.

**Discussion:**  The true potential of our reformulation is in dealing with more structured norms rather than $\ell_1$, where computing the proximal operator is often intractable. One such case is the latent group lasso norm [46], defined as

$$
\|z\|_\Omega = \sum_{i=1}^{I} \|z_{\Omega_i}\|,
$$

Figure 2: Basis Pursuit

where $\{\Omega_i\}_{i=1}^I$ are (not necessarily disjoint) index sets of $\{1, \cdots, d\}$. Although not studied here, we believe that the nonconvex framework presented in this paper can serve to solve more complicated problems, such as the latent group lasso. We leave this research direction for future work.

**Condition verification:**  In the sequel, we verify that Theorem 4.1 indeed applies to (1) with the above $f, A, g$. Note that

$$DA(x) = 2\overline{B}\text{diag}(x), \tag{62}$$

where $\text{diag}(x) \in \mathbb{R}^{2d \times 2d}$ is the diagonal matrix formed by $x$. The left-hand side of (18) then reads as

$$
\begin{aligned}
\text{dist}&\left(-DA(x_k)^\top A(x_k), \frac{\partial g(x_k)}{\beta_{k-1}}\right)\\
&= \text{dist}\left(-DA(x_k)^\top A(x_k), \{0\}\right) \qquad (g \equiv 0)\\
&= \|DA(x_k)^\top A(x_k)\|\\
&= 2\|\text{diag}(x_k)\overline{B}^\top (\overline{B}x_k^{\circ 2} - b)\|. \qquad \text{(see (62))}
\end{aligned} \tag{63}
$$

To bound the last line above, let $x_*$ be a solution of (1) and note that $\overline{B}x_*^{\circ 2} = b$ by definition. Let also $z_k, z_* \in \mathbb{R}^d$ denote the vectors corresponding to $x_k, x_*$. Corresponding to $x_k$, also define $u_{k,1}, u_{k,2}$ naturally and let $|z_k| = u_{k,1}^{\circ 2} + u_{k,2}^{\circ 2} \in \mathbb{R}^d$ be the vector of amplitudes of $z_k$. To simplify matters, let us assume also that $B$ is full-rank. We then rewrite the norm in the last line of (63) as

$$
\begin{aligned}
\|\text{diag}&(x_k)\overline{B}^\top (\overline{B}x_k^{\circ 2} - b)\|^2\\
&= \|\text{diag}(x_k)\overline{B}^\top \overline{B}(x_k^{\circ 2} - x_*^{\circ 2})\|^2 \qquad (\overline{B}x_*^{\circ 2} = b)\\
&= \|\text{diag}(x_k)\overline{B}^\top B(x_k - x_*)\|^2\\
&= \|\text{diag}(u_{k,1})B^\top B(z_k - z_*)\|^2\\
&\quad + \|\text{diag}(u_{k,2})B^\top B(z_k - z_*)\|^2\\
&= \|\text{diag}(u_{k,1}^{\circ 2} + u_{k,2}^{\circ 2})B^\top B(z_k - z_*)\|^2\\
&= \|\text{diag}(|z_k|)B^\top B(z_k - z_*)\|^2\\
&\geq \eta_n(B\text{diag}(|z_k|))^2 \|B(z_k - z_*)\|^2\\
&= \eta_n(B\text{diag}(|z_k|))^2 \|Bz_k - b\|^2 \qquad (Bz_* = \overline{B}x_*^{\circ 2} = b)\\
&\geq \min_{|T|=n} \eta_n(B_T) \cdot |z_{k,(n)}|^2 \|Bz_k - b\|^2,
\end{aligned} \tag{64}
$$

where $\eta_n(\cdot)$ returns the $n$th largest singular value of its argument. In the last line above, $B_T$ is the restriction of $B$ to the columns indexed by $T$ of size $n$. Moreover, $z_{k,(n)}$ is the $n$th largest entry of $z$ in magnitude. Given a prescribed $\nu$, (18) therefore holds if

$$|z_{k,(n)}| \geq \frac{\nu}{2\sqrt{\min_{|T|=n} \eta_n(B_T)}}, \tag{65}$$

for every iteration $k$. If Algorithm 1 is initialized close enough to the solution $z^*$ and the entries of $z^*$ are sufficiently large in magnitude, there will be no need to directly enforce (65).

## E.2 Generalized Eigenvalue Problem

Figure 3: *(Top)* Objective convergence for calculating top generalized eigenvalue and eigenvector of $B$ and $C$. *(Bottom)* Eigenvalue structure of the matrices. For (i),(ii) and (iii), $C$ is positive semidefinite; for (iv), (v) and (vi), $C$ contains negative eigenvalues. [(i): Generated by taking symmetric part of iid Gaussian matrix. (ii): Generated by randomly rotating $\mathrm{diag}(1^{-p}, 2^{-p}, \cdots, 1000^{-p})(p = 1)$. (iii): Generated by randomly rotating $\mathrm{diag}(10^{-p}, 10^{-2p}, \cdots, 10^{-1000p})(p = 0.0025)$.]

Generalized eigenvalue problem has extensive applications in machine learning, statistics and data analysis [26]. The well-known nonconvex formulation of the problem is [13] given by

$$\begin{cases} \min_{x \in \mathbb{R}^n} x^\top C x \\ x^\top B x = 1, \end{cases} \tag{66}$$

where $B, C \in \mathbb{R}^{n \times n}$ are symmetric matrices and $B$ is positive definite, namely, $B \succ 0$. The generalized eigenvector computation is equivalent to performing principal component analysis (PCA) of $C$ in the norm $B$. It is also equivalent to computing the top eigenvector of symmetric matrix $S = B^{-1/2} C B^{1/2}$ and multiplying the resulting vector by $B^{-1/2}$. However, for large values of $n$, computing $B^{-1/2}$ is extremely expensive. The natural convex SDP relaxation for (66) involves lifting $Y = xx^\top$ and removing the nonconvex $\mathrm{rank}(Y) = 1$ constraint, namely,

$$\begin{cases} \min_{Y \in \mathbb{R}^{n \times n}} \mathrm{tr}(CY) \\ \mathrm{tr}(BY) = 1, \quad X \succeq 0. \end{cases} \tag{67}$$

Here, however, we opt to directly solve (66) because it fits into our template with

$$f(x) = x^\top C x, \quad g(x) = 0,$$
$$A(x) = x^\top B x - 1. \tag{68}$$

We compare our approach against three different methods: manifold based Riemannian gradient descent and Riemannian trust region methods in [11] and the linear system solver in [26], abbrevated as GenELin. We have used Manopt software package in [12] for the manifold based methods. For GenELin, we have utilized Matlab's backslash operator as the linear solver. The results are compiled in Figure 3.

**Condition verification:** Here, we verify the regularity condition in (18) for problem (66). Note that

$$DA(x) = (2Bx)^\top. \tag{69}$$

Therefore,

$$\operatorname{dist}\left(-DA(x_k)^\top A(x_k), \frac{\partial g(x_k)}{\beta_{k-1}}\right)^2 = \operatorname{dist}\left(-DA(x_k)^\top A(x_k), \{0\}\right)^2 \qquad (g \equiv 0)$$
$$= \|DA(x_k)^\top A(x_k)\|^2$$
$$= \|2Bx_k(x_k^\top B x_k - 1)\|^2 \qquad \text{(see (69))}$$
$$= 4(x_k^\top B x_k - 1)^2 \|Bx_k\|^2$$
$$= 4\|Bx_k\|^2 \|A(x_k)\|^2 \qquad \text{(see (68))}$$
$$\geq \eta_{\min}(B)^2 \|x_k\|^2 \|A(x_k)\|^2, \tag{70}$$

where $\eta_{\min}(B)$ is the smallest eigenvalue of the positive definite matrix $B$. Therefore, for a prescribed $\nu$, the regularity condition in (18) holds with $\|x_k\| \geq \nu/\eta_{min}$ for every $k$. If the algorithm is initialized close enough to the constraint set, there will be again no need to directly enforce this latter condition.

### E.3 $\ell_\infty$ Denoising with a Generative Prior

The authors of [56, 30] have proposed to project onto the range of a Generative Adversarial network (GAN) [28], as a way to defend against adversarial examples. For a given noisy observation $x^* + \eta$, they consider a projection in the $\ell_2$ norm. We instead propose to use our augmented Lagrangian method to denoise in the $\ell_\infty$ norm, a much harder task:

$$\begin{aligned} \min_{x,z} \quad & \|x^* + \eta - x\|_\infty \\ \text{s.t.} \quad & x = G(z). \end{aligned} \tag{71}$$

Figure 4: Augmented Lagrangian vs Adam and Gradient descent for $\ell_\infty$ denoising

We use a pretrained generator for the MNIST dataset, given by a standard deconvolutional neural network architecture [52]. We compare the succesful optimizer Adam [34] and gradient Descent

against our method. Our algorithm involves two forward and one backward pass through the network, as oposed to Adam that requires only one forward/backward pass. For this reason we let our algorithm run for 2000 iterations, and Adam and GD for 3000 iterations. Both Adam and gradient descent generate a sequence of feasible iterates $x_t = G(z_t)$. For this reason we plot the objective evaluated at the point $G(z_t)$ vs iteration count in figure 4. Our method successfully minimizes the objective value, while Adam and GD do not.

### E.4 Quadratic assginment problem

Let $K$, $L$ be $n \times n$ symmetric metrices. QAP in its simplest form can be written as

$$\max \operatorname{tr}(KPLP), \quad \text{subject to } P \text{ be a permutation matrix} \tag{72}$$

A direct approach for solving (72) involves a combinatorial search. To get the SDP relaxation of (72), we will first lift the QAP to a problem involving a larger matrix. Observe that the objective function takes the form

$$\operatorname{tr}((K \otimes L)(\operatorname{vec}(P)\operatorname{vec}(P^\top))),$$

where $\otimes$ denotes the Kronecker product. Therefore, we can recast (72) as

$$\operatorname{tr}((K \otimes L)Y) \quad \text{subject to } Y = \operatorname{vec}(P)\operatorname{vec}(P^\top), \tag{73}$$

where $P$ is a permutation matrix. We can relax the equality constraint in (73) to a semidefinite constraint and write it in an equivalent form as

$$X = \begin{bmatrix} 1 & \operatorname{vec}(P)^\top \\ \operatorname{vec}(P) & Y \end{bmatrix} \succeq 0 \text{ for a symmetric} X \in \mathbb{S}^{(n^2+1)\times(n^2+1)}$$

We now introduce the following constraints such that

$$B_k(X) = \mathbf{b_k}, \quad \mathbf{b_k} \in \mathbb{R}^{m_k} \tag{74}$$

to make sure X has a proper structure. Here, $B_k$ is a linear operator on $X$ and the total number of constraints is $m = \sum_k m_k$. Hence, SDP relaxation of the quadratic assignment problem takes the form,

$$\begin{aligned}
\max \quad & \langle C, X \rangle \\
\text{subject to } \quad & P1 = 1, \ 1^\top P = 1, \ P \geq 0 \\
& \operatorname{trace}_1(Y) = I \ \operatorname{trace}_2(Y) = I \\
& \operatorname{vec}(P) = \operatorname{diag}(Y) \\
& \operatorname{trace}(Y) = n \begin{bmatrix} 1 & \operatorname{vec}(P)^\top \\ \operatorname{vec}(P) & Y \end{bmatrix} \succeq 0,
\end{aligned} \tag{75}$$

where $\operatorname{trace}_1(.)$ and $\operatorname{trace}_2(.)$ are partial traces satisfying,

$$\operatorname{trace}_1(K \otimes L) = \operatorname{trace}(K)L \quad \text{and} \quad \operatorname{trace}_2(K \otimes L) = K\operatorname{trace}(L)$$

$$\operatorname{trace}_1^*(T) = I \otimes T \quad \text{and} \quad \operatorname{trace}_2^*(T) = T \otimes I$$

$1st$ set of equalities are due to the fact that permutation matrices are doubly stochastic. $2nd$ set of equalities are to ensure permutation matrices are orthogonal, i.e., $PP^\top = P^\top P = I$. $3rd$ set of equalities are to enforce every individual entry of the permutation matrix takes either 0 or 1, i.e., $X_{1,i} = X_{i,i} \ \forall i \in [1, n^2+1].$. Trace constraint in the last line is to bound the problem domain. By concatenating the $B_k$'s in (74), we can rewrite (75) in standard SDP form as

| Data | Optimal Value | Sparse QAP [23] | Optimality Gap (%) | | | | |
|------|---------------|-----------------|-------------------|---|---|---|---|
| | | | iAL | | | | |
| | | | $r = 10$ | $r = 25$ | $r = 50$ | $r = r_m$ | $r_m$ |
| esc16a | 68 | 8.8 | 11.8 | **0** | **0** | 5.9 | 157 |
| esc16b | 292 | **0** | **0** | **0** | **0** | **0** | 224 |
| esc16c | 160 | 5 | 5.0 | 5.0 | **2.5** | 3.8 | 177 |
| esc16d | 16 | 12.5 | 37.5 | **0** | **0** | 25.0 | 126 |
| esc16e | 28 | 7.1 | 7.1 | **0** | 14.3 | 7.1 | 126 |
| esc16g | 26 | **0** | 23.1 | 7.7 | **0** | **0** | 126 |
| esc16h | 996 | **0** | **0** | **0** | **0** | **0** | 224 |
| esc16i | 14 | **0** | **0** | **0** | 14.3 | **0** | 113 |
| esc16j | 8 | **0** | **0** | **0** | **0** | **0** | 106 |
| esc32a | 130 | 93.8 | 129.2 | 109.2 | 104.6 | **83.1** | 433 |
| esc32b | 168 | 88.1 | 111.9 | 92.9 | 97.6 | **69.0** | 508 |
| esc32c | 642 | 7.8 | 15.6 | 14.0 | 15.0 | **4.0** | 552 |
| esc32d | 200 | 21 | 28.0 | 28.0 | 29.0 | **17.0** | 470 |
| esc32e | 2 | **0** | **0** | **0** | **0** | **0** | 220 |
| esc32g | 6 | **0** | 33.3 | **0** | **0** | **0** | 234 |
| esc32h | 438 | 18.3 | 25.1 | 19.6 | 25.1 | **13.2** | 570 |
| esc64a | 116 | 53.4 | 62.1 | 51.7 | 58.6 | **34.5** | 899 |
| esc128 | 64 | **175** | 256.3 | 193.8 | 243.8 | 215.6 | 2045 |

Table 1: Comparison between upper bounds on the problems from the QAP library with (relatively) sparse $L$.

$$
\begin{aligned}
\max \quad & \langle C, X \rangle \\
\text{subject to} \quad & B(X) = \mathbf{b}, \ \ \mathbf{b} \in \mathbb{R}^m \\
& \text{trace}(X) = n + 1 \\
& X_{ij} \geq 0, \ \ i, j \ \mathcal{G} \\
& X \succeq 0,
\end{aligned}
\tag{76}
$$

where $\mathcal{G}$ represents the index set for which we introduce the nonnegativities. When $\mathcal{G}$ covers the wholes set of indices, we get the best approximation to the original problem. However, it becomes computationally undesirable as the problem dimension increases. Hence, we remove the redundant nonnegativity constraints and enforce it for the indices where Kronecker product between $K$ and $L$ is nonzero.

We penalize the non-negativity constraints and add it to the augmented Lagrangian objective since a projection to the positive orthant approach in the low rank space as we did for the clustering does not work here.

We take [23] as the baseline. This is an SDP based approach for solving QAP problems containing a sparse graph. We compare against the best feasible upper bounds reported in [23] for the given instances. Here, optimality gap is defined as

$$
\%\text{Gap} = \frac{|\text{bound} - \text{optimal}|}{\text{optimal}} \times 100
$$

We used a (relatively) sparse graph data set from the QAP library. We run our low rank algorithm for different rank values. $r_m$ in each instance corresponds to the smallest integer satisfying the Pataki bound [49, 1]. Results are shown in Table 1. Primal feasibility values except for the last instance $esc128$ is less than $10^{-5}$ and we obtained bounds at least as good as the ones reported in [23] for these problems.

For $esc128$, the primal feasibility is $\approx 10^{-1}$, hence, we could not manage to obtain a good optimality gap.