[Reviews · NeurIPS 2019]

Reviewer 1



The authors propose to solve a non-convex optimization problem with non-linear constraints, which is a common template for a variety of problems in machine learning. The authors solve the primal problem inexactly, controlled by error \eps_k, which is gradually decreased, as penalty \beta_k is increased and the iterates approach stationarity. The inexact problem is solved with the help of both first order and second order solvers and this paper analyzes overall computational complexity of iALM under both kinds of substitutions, which is a first. The paper is well written, with good references and presentation. The experimental analysis gives useful insights into the performances of iALM with both Accelerated Proximal Gradient Method, a first order approach and 1BFGS.

Reviewer 2



Applying ALM to the Burer-Monterio problem and to nonlinear programs in general is natural and well summarized in the monograph Ref [8]. Allowing first-order and second-order approximate solvers for the primal subproblems is also classic, and can be found in, e.g., Ch 8 & 9 of Ref [8]. I think the main novelties here lie at the nonsmooth, convex term g(x) and the convergence rate results. Sec 5 of the paper has provided a comprehensive review of pertinent results under different assumptions. I have several concerns that I hope the authors can address: * The BM example does not quite justify the inclusion of the possibly nonsmooth term g in (1). The authors may want to balance out and briefly discuss other examples as appearing in the experiments. * Suppose there is no g, are there already existing convergence results of similar nature? Also, would the idea of proof be substantially different? If possible, it may help to sketch the high level idea of the proof in the main text. * Compared to the convex case, Alg 1 for the nonconvex case is obviously more complicated. I think for the benefit of the machine learning community which are mostly only familiar with the convex case, the authors may want to explain the main changes, e.g., why \sigma_k needs to be updated. * For the presentation in Sec 5, would it be better to tabulate the main results rather than the current form? It cost a bit of effort to really digest the fragmented pieces. * Maybe I'm missing something---the paper has mentioned twice that the regularity condition in (15) will be checked in Sec 6, i.e., the experiment. But I've not detected the presence. * I was reminded of a possible dual submission of this paper with paper # 6483, which I also reviewed. It seems to me that the linearized ADMM can indeed also be applied to the current model problem (1), although I'm not sure if a convergence rate result can be obtained. On the reverse direction, it's not clear if one can apply modified version of inexact ALM to the problem in # 6483 can still obtain the global results therein. Maybe the authors want to clarify on these.

Reviewer 3



I have read the rebuttal and some of my concerns are raised. I admit that I have missed the verification for (15) for 3 specific cases in the appendix in E&F, mostly since in the original paper it incorrectly says it will be mentioned in section 6. But anyway overall the paper has some values in the convergence analysis of some specific problems including clustering etc. Therefore I have modified my score. However, I still think the contribution of this paper is overstated and the statements for corollary 4.2/4.3 are not entirely valid. Normally for optimization results, given the function f, and g this work is targeting to optimize upon, the assumption should only rely on the geometry of f and g. Therefore one could tell if the functions satisfy this assumption or not, and whether the algorithm would work. Of course it could also include some tolerance on the optimality of the primal progress as the equation in step 2 of algorithm 1 does. It is valid since given enough optimization steps, it could be achieved. However, equation (15) is some contrived condition that depends on the intermediate x_k and the beta_k that is not clearly what it means *in general*. I agree that the author have discussed and verified (15) for 3 cases all happened *when g=0 or is an indicator function*. And in this scenario, the function almost has nothing to do with g and it only relies on the geometry of f on the valid domain, which is fine. But clearly the paper is overstating the contribution for their conclusion for general convex g. Plus the assumption is simply omitted in the corrollary 4.2/4.3 without verification. This makes it looks like they show such convergence rate for general f and g, which is not correct. I'm fine with accepting the paper, but at least the authors should improve the presentation. Please move some of the valuable results from the appendix to the main text, and modify the corollaries so that they're self-content with the proper assumptions included. ======================================================== 1. My biggest concern is that the theoretical guarantee is not at all rigorous. It uses a weird and contrived assumption (15), assuming the function satisfies some property with parameter beta_k that depends on the number of iteration k. Normally a proper assumption should be only on the geometry of the loss, and be independent with the intermediate iterates. Meanwhile, no explanation is provided for this beta_k. Later the authors simply assume beta_k to be some b^k, still without explainations. For such assumptions, the authors have not even shown any real problems that meet the requirements. 2. The major content of the paper is about literature review. For the real contribution part, it fails to provide any intuition on how the method works, and why does it propose such a weird assumption. 3. The mathematical language is not compact and weird. The theorems are written in a sloppy way, with some key terms only defined in the appendix. 4. Finally, the experimental results are not convincing either. The only example in the main text shows improved efficiency on a single dataset for a single task, while all other experiments in the appendix show negative results.

[Author Response · NeurIPS 2019]

We thank the reviewers for their time and feedback. To our knowledge, our work provides the first *practical* algorithm with *provable* iteration complexity for solving nonconvex optimization problems with nonlinear constraints which has numerous applications in machine learning. We argue that the proposed framework will be a staple for many such applications in the future, and our theory-based code will be made public, which will have sustained impact.

We first address the concerns of R3 on condition (15) and we wish R3 would reconsider their score:

1. To our knowledge, our condition (15) is new and inspired by our theoretical analysis. Mangasarian-Fromovitz constraint qualification, that is commonly assumed in the literature, is assumed on a specific region of the space, whereas our condition is algorithm dependent which we believe to be a weaker requirement. This is precisely the reason why the iteration count appears in the condition. We find this to be a strength rather than a weakness.

2. On the relation of (15) and $\beta_k$: We consider two cases, when $g$ is the indicator of a convex set (or 0), the subdifferential set will be a cone (or 0), thus $\beta_k$ will not have an effect. On the other hand, when $g$ is a convex function defined on the whole space (please see Thm.3.1.13 of Nesterov's book), subdifferential set will be bounded. This will introduce an error term in (15) that is of the order $(1/\beta_k)$. One can see that $b^k$ choice for $\beta_k$ causes a linear decrease in this error term. In fact, all the examples in this paper fall into the first case. However, for generality, we will clarify these two cases in the main text.

3. The specific choice of $\beta_k = b^k$ was motivated by practical performance, and as argued above, compatible with (15).

4. We in fact validate (15) in Appendices E and F for some key examples. In the convex case, the geometric variant of (15) holds iff the Slater's condition holds (under additional assumptions).

5. The condition (15) is discussed in detail in the 'Regularity' paragraph in Section 4 with its connections to the existing conditions. We will add more intuition; however, we have been quite exhaustive in our literature review and strongly believe in the utility of (15) in the future.

Moreover, we respectfully disagree with R3 on the negativity of the numerical results. In fact,

1. We apply our algorithm to five diverse applications, which by itself is notable for a theoretical submission. In all cases, we perform head-to-head or better than the highly tailored algorithms for each application. These algorithms often do not have the generality and flexibility of our framework, which cannot be understated. Please see for instance the discussion in supplementary material, starting line 578.

2. We verify condition (15) for three key examples; clustering, basis pursuit and generalized eigenvalue decomposition.

3. We propose a nonconvex formulation of the standard basis pursuit template which can handle non-seperable priors (e.g., structured norms such as those arising in group Lasso) for which the baselines methods are not applicable. Please refer to 'discussion' paragraph in Appendix E.1.

4. We provide new state-of-the-art results for QAP, which is considered a difficult problem. We perform at least as good as the baseline for 18 datasets out of 19.

R2's double submission remark: We will clarify the differences in the camera ready. Specifically, 6483 considers strongly convex optimization subject to nonlinear constraints with an ADMM framework different from our general setting of nonconvex optimization and ALM instead of ADMM. The assumptions required for nonlinear operators are also different. In short, our submission is solving a more general framework than 6483, with a different algorithm. This requires different analysis and substantially different results. Moreover, results also do not reduce to each other in the specific cases.

Response to the remaining comments are in the sequel.

- **R1** & **R2:** As suggested by R1 and R2, we will summarize our literature review as a table for improved readability.

- **R2:** In clustering in section 6, $g \neq 0$ is the indicator function of a convex set, which justifies the BM example.

- **R2:** When $g = 0$, similar theoretical guarantees with more complicated (not simple to implement) algorithms are obtained in [7, 18]. Our proof idea is similar in the $g = 0$ case. We will include the idea of the proof in the main text.

- **R2:** The particular choice of $\sigma_k$ is to keep the dual sequence bounded (see line 98). We will add more discussion for the differences in the nonconvex case.

- **R3:** Please see above for our clarification involving Condition (15).

- **R3:** The method we analyze is the classical ALM, the intuition for which is well-understood, and we will highlight this briefly in the final version. Condition (15) stems from the analysis, and carefully described in the text.

- **R3:** The technical results are self-contained, except $y_{\max}$, for which we had given a pointer to its expression for the sake of space. We will move back the expression for $y_{\max}$ to the text.

- **R3:** Please see above for practical remarks. We ran additional experiment for clustering on MNIST digits (in addition to fashion-MNIST) and converged at least 5 times faster with both apgm and lbfgs solvers. We can include these results in the camera ready.

[Meta-Review · NeurIPS 2019]

This paper uses the Augmented Lagrangian method to solve optimization problems for a sum of functions f and g, where f is nonconvex and g is convex but 'proximal-friendly' subject to quite general nonlinear constraints. The proposed method solves the primal problem within some error epsilon_k that is gradually decreased as a penalty schedule beta_k is increasing across iterations. The approximate intermediate problems are solved using first order and second order solvers. The proposed analysis is technically non-trivial and interesting. The presentation of the paper was poor and at times confusing which made this a borderline paper. The algorithmic and convergence results are novel, but the applicability was not fully understood. This is because of the assumptions that the authors require for their framework and convergence analysis to work. Specifically, after discussions and considering the rebuttal of the authors, R4 is still concerned that the assumption (15) is strong. Further, it was confusing to us that the assumption depends on intermediate iterates of the proposed algorithm. We understand that the authors introduce beta_k to provide some slackness in the primal updates and the conditions like the equation in Algorithm 1, step 2 is valid. This could be achieved when the oracle is called sufficiently many times. The problem is that equation (15) is an assumption that one needs to verify for a specific problem. Gladly, the authors did indeed verify it for the cases when function g does not play a role (in appendix E/F), but not for a general function g (or at least an interesting sub-family). In Appendix B, they try to prove corollary 4.2, but they actually did not verify this assumption (15). As far as we understand, in appendix B the authors assume (15) is true (for some unspecified g), and just use the conclusion of Thm 4.1 to show that they could converge to the (epsilon_f,beta_k) approximation. However, in the statements of corollaries 4.2/4,3, this assumption (15) is not explicitly stated, which was confusing. The authors should discuss this point in more detail and make sure all the assumptions are clearly stated in the statement of every theorem or corollary, so that the reader does not have to dig into an appednix to understand how to use a corollary. In conclusion, we are sure there are some families of functions for which the claimed results hold. There are also some statements in the paper that we are not sure if they hold unless specific technical assumptions are true, but they are clarified in the appendices. Still, after discussions, we believe that the mathematical results that are clear (i.e. when g does not play a role) are still sufficiently interesting for publication as a poster. We strongly encourage the authors to clarify these technical issues in the final version of the paper..